# Spatial Warped Gaussian Processes: Estimation and Efficient Field Reconstruction

**DOI:** 10.3390/e23101323

**Published:** 2021-10-11

**Authors:** Gareth W. Peters, Ido Nevat, Sai Ganesh Nagarajan, Tomoko Matsui

**Affiliations:** 1Department of Statistics and Applied Probability, University of California Santa Barbara, Santa Barbara, CA 93106, USA; 2TUMCREATE, 1 Create Way, #10-02 CREATE Tower, Singapore 138602, Singapore; ido.nevat@tum-create.edu.sg; 3Engineering Systems and Design Pillar, Singapore University of Technology and Design (SUTD), 8 Somapah Road, Singapore 487372, Singapore; sai.nagarajan@epfl.ch; 4Department of Statistical Modeling, The Institute of Statistical Mathematics (ISM), 10-3 Midoricho, Tachikawa, Tokyo 190-0014, Japan; tmatsui@ism.ac.jp

**Keywords:** random fields, warped Gaussian process, spatial field reconstruction

## Abstract

A class of models for non-Gaussian spatial random fields is explored for spatial field reconstruction in environmental and sensor network monitoring. The family of models explored utilises a class of transformation functions known as Tukey g-and-h transformations to create a family of warped spatial Gaussian process models which can support various desirable features such as flexible marginal distributions, which can be skewed, leptokurtic and/or heavy-tailed. The resulting model is widely applicable in a range of spatial field reconstruction applications. To utilise the model in applications in practice, it is important to carefully characterise the statistical properties of the Tukey g-and-h random fields. In this work, we study both the properties of the resulting warped Gaussian processes as well as using the characterising statistical properties of the warped processes to obtain flexible spatial field reconstructions. In this regard we derive five different estimators for various important quantities often considered in spatial field reconstruction problems. These include the multi-point Minimum Mean Squared Error (MMSE) estimators, the multi-point Maximum A-Posteriori (MAP) estimators, an efficient class of multi-point linear estimators based on the Spatial-Best Linear Unbiased (S-BLUE) estimators, and two multi-point threshold exceedance based estimators, namely the Spatial Regional and Level Exceedance estimators. Simulation results and real data examples show the benefits of using the Tukey g-and-h transformation as opposed to standard Gaussian spatial random fields in a real data application for environmental monitoring.

## 1. Introduction

Spatial monitoring of physical phenomena has been at the center of many different spatial statistics and signal processing innovations in recent decades. Specific applications in which environmental processes are monitored and spatial field reconstruction is performed include environmental monitoring [1] and weather monitoring [2,3,4], natural phenomena such as temperature [5], precipitation [6], wind speed [7,8,9,10], pollution [11] and signal processing [12,13] to name a few.

In this work, we explore warping maps of Gaussian process models such as those developed in [7] for spatial field reconstruction. It is most commonly the case that spatial field modelling is undertaken with a variation of a Gaussian Process (GP) model due to its direct interpretation and mathematical tractability [14]. GP models are readily characterized by a mean structure and a valid covariance function [14]. Such models have been applied in many different domains for spatial and temporal modelling: [15,16,17] and the classic texts on such Gaussian process models for spatial statistics [18,19]. The warping extensions studied in this work are designed to specifically accommodate a flexible range of non-Gaussian skew and kurtosis attributes in order to demonstrate how to undertake accurate and efficient spatial field reconstruction in a more general class of spatial process models, whilst still maintaining much of the practical interpretation offered by Gaussian process modelling settings.

The significance of adding such warping structure arises since increasingly spatial modellers and data scientists are observing that many types of physical phenomena may have finite dimensional distributions which are not adequately captured by symmetric light tailed Gaussian distributions as is mandated by such classical GP spatial field modelling approaches. The reality is that the skew and kurtosis can play a significant role in spatial field modelling, see discussion in [4,20,21]. As noted in these works, there are many real world settings where the physical process being modelled in the spatial field reconstruction may exhibit features which cannot be captured accurately via GP modelling, as the process may display spatial features at different locations that include excess skewness (asymmetry about the mode), kurtosis and perhaps heavy tails. Hence, the GP will not provide such flexible features if used as the basis upon which to undertake development of estimators for spatial field reconstruction, therefore we wish to extend this class of models. This will in turn influence the accuracy of any spatial field reconstruction estimator built from such statistical model assumptions.

Therefore, when one finds a setting in which GP structure will not adequately capture the spatial field statistical structure, there are numerous approaches that one may proceed with when seeking to perform spatial field reconstruction. The first involves seeking a transformation that will induce properties in the transformed observations of the process more akin to a GP model, see examples in [22,23] for a detailed overview of the subject. Examples include the log-transform, proposed by Galton, which resulted in the nowadays well-known log-normal distribution [24]. The first systematic attempt to fit asymmetric distributions to non-normal data was made by Edgeworth [25]. Karl Pearson defined several probability distributions as solutions to a particular differential equation, which formed the basis for the Pearson family of distributions [26]. Further development was the two-piece distributions, proposed by Fechner, which were developed by binding together two halves of normal curves, each having a different scaling [27]. A more recent work which had a great impact was by Azzalini [28] who provided a rigorous treatment to the analysis of Skew-Normal distributions.

An alternative, recent class of approaches in the statistical machine learning literature has involved the converse of these methods, where one rather warps or “transforms” a GP model to produce an additional structure such as skewness or kurtosis and then works with estimation and spatial field reconstruction of the resultant transformed process. This method has two components—the specification of the sufficient statistical functions for the input GP model and the specification of the functional characteristic of the transformation map.

In this work, we explore the second approach based on warping a GP model. One may consider our approach as exploring a generalised class of processes that act as extensions of the GP family to a much larger family of processes that incorporate parameterised features of skewness and kurtosis that arises directly from the class of warping functions adopted. In this family of models we develop, just like the GP case, we wish to be able to state useful information about the resulting processes finite dimensional distributions: i.e., the marginal and conditional distributions of the process at different locations in space. This can be done by modifying an existing known parametric distribution (Gaussian) by applying a transformation of the random variables pointwise. This concept was first proposed by Tukey in 1977, and further developed in [29,30,31] and in the time series setting by [32]. In this paper, we demonstrate that when one selects the family of Tukey g-and-h model distortion functions, one can derive tractable solutions to spatial field reconstruction estimators that will out-perform classical GP solutions when spatial processes exhibit excess spatial skewness and spatial kurtosis features.

### Contributions and Outline

The main contributions developed in this work are twofold: formulating a class of spatial warped GP models based on Tukey g-and-h warping functions. This involves deriving important key properties of the Tukey g-and-h transformation and resulting family of spatial warped GP (Section 2 and Section 3); and using those results to develop new estimation algorithms for efficient spatial field reconstructions (Section 4). Our contributions may be summarised as follows:(1)In Section 2, we derive mathematical properties of the proposed warped GP models warping function known as the Tukey g-and-h transformation function. This is actually a class of warping functions with sub-classes related to skewness and kurtosis transformations. Since this transformation is not widely known, it is valuable for practitioners to see the basic properties of this class of functions in order to better understand the resulting properties of the warped GP that arise when using this transform. In this regard we explore the transformation and its limiting behaviour (Proposition 2), the existence of the inverse transform (Proposition 1), and its derivatives (Lemma 1). This set of results is instrumental for the analysis of the properties of the Tukey g-and-h process and for the construction of spatial field approximations that form the key application in this paper.(2)In Section 3, we derive important statistical properties of the warped GP process which include properties of the finite dimensional distributions such as raw and central moments, the multivariate moments (Corollary 1), the population mean and variance (Corollaries 3–4), its higher order multivariate cross moments of any order (Proposition 3), and the index of regular variation (Lemma 2), to determine conditions in terms of the parameters of the g-and-h transform for the existence of all higher order mixed moments. In particular, the cross moments are directly relevant to the construction of efficient spatial field reconstructions for multiple locations in space that are performed in the application section. The tail behaviour of the finite dimensional distributions is captured by the tail index analysis and is relevant to understand models with excess kurtosis relative to a GP model, which is induced by our class of warping function.(3)In Section 4, we derive the spatial field reconstruction estimators of the Tukey g- and-h random field under five different metrics, most of which are based on the predictive posterior density (Lemma 4).(4)In Section 5, a detailed set of numerical experiments are provided. This includes both real data cases studies as well as synthetic case studies which were developed to demonstrate key properties of the spatial field reconstruction methods and their accuracy and performance relative to a baseline GP model. In regards to the real data case studies, a sequence of experiments was developed which was based on data obtained from the National Climatic Data Center (NCDC) which included spatial observations of precipitation obtained from ground monitoring stations in the regions of interest. In particular, we took one month of hourly precipitation measurements to fit our warped GP model to obtain a spatial field reconstruction which was compared to a baseline GP model with no warping in order to demonstrate the benefits of a spatial warping framework.

## 2. Constructing a Flexible Warping Function for Spatial Gaussian Processes

Many families of warping function can be developed for extending the class of GP spatial process models. In this work, we particularly focus on the study of the family of Tukey g-and-h warping functions. When developing the new warped GP class of flexible models, the following list of desirable properties was considered desirable for the resulting warped GP model to posses (see [22]):(1)Parsimonious and finite number of well interpreted parameters;(2)Well-separated roles for skewness, kurtosis and the tail behaviour parameters;(3)Accurate and computationally efficient parameter estimation and tractability;(4)Derivable and interpretable stochastic properties;(5)Flexible structures in the marginal, conditional and joint density shapes;(6)Good inferential properties;(7)Exact and computationally efficient data generating mechanisms.

Some of these aforementioned properties have been studied before in the context of the class of warping functions we consider, namely the Tukey g-and-h transformed random GP fields, see for example [31,32,33,34,35,36] and the textbook overview in [37].

In this paper, we re-explore these properties and extend them for the purposes of spatial field reconstruction. Therefore, in this section, we first define the fundamental building blocks for deriving the non-Gaussian warped GP fields. As such, we define the basic set of non-linear transformations of univariate and multivariate Gaussian random variables and the resulting Tukey g-and-h univariate and multivariate distributions. We also derive some fundamental properties such as the existence of the inverse transform as well as the derivatives of the distribution that will be important for future results.

**Definition** **1**(Tukey Transformation [38])**.**
*Consider a Gaussian random variable $W∼N\left(\mu ,\sigma \right)$ and transformation $Y:=T\left(W;ı\right)$ with parameter vector $ı:=\left[m_{0},m_{1},\mu ,\sigma^{2},\eta ,\theta \right]$, where $m_{0}\in R$, $m_{1}\in R^{+}$, $\eta \in R^{d}$ and θ∈R. Then the resultant transformed random variable Y will be from a Tukey law if the corresponding transformation $T\left(W;\zeta \right)$ is given by*
(1)$$\begin{array}{c} Y:=T\left(W;\zeta \right)=m_{0}+m_{1}{\left(F\left(W;\eta \right)\right)}^{\theta }W, \end{array}$$
*where the transformation $F\left(W;\eta \right)$ is positive, symmetric, and strictly monotonically increasing for positive values of W≥0.*

We note that, as a default, we will consider the base transformed random variable *W* to be Gaussian with μ=0 and σ=1 unless otherwise stated. Furthermore, in general the transformation $F\left(W;\eta \right)$ can have many forms, see discussions in [32]. In this manuscript we concentrate on two specific types of transformation directly related to inducing skewness in the warped GP model or inducing kurtosis in the warped GP model. Note that standard GP models can have flexible covariance functions (second order flexibility) but there is no flexibility in the higher order structure relating to third co-moments (co-skewness) and fourth co-moments (co-kurtosis) that can often be observed in real spatial field observation data. Therefore, the use of the Tukey sub-families of warping functions that specifically produce parameterization of such structure in the warped GP model is of practical relevance in flexible reconstruction methods for spatial fields.

**Definition** **2**(Tukey Family of Transformations)**.**
*The Tukey g (skewness) and Tukey h (kurtosis) transformations form two classes of sub-transformations in the Tukey family which are characterised by warping functions given by:*

*(1)* 
*Skew transform:*

(2)
$$\begin{array}{c} \begin{array}{cc} G\left(W;g\right) & =\frac{\exp\left(gW\right)-1}{gW}. \end{array} \end{array}$$

*(2)* 
*Kurtosis transform:*

(3)
$$\begin{array}{c} \begin{array}{cc} H\left(W;h\right) & =\exp\left(\frac{hW^{2}}{2}\right), \end{array} \end{array}$$

*(3)* 
*Skew-Kurtosis product transform:*

(4)
$$\begin{array}{cc} F\left(W;\eta \right) & =G\left(W;g\right)H\left(W;h\right), \end{array}$$


*for $g\in R-\left\{0\right\}$, $h\in R^{+}$ and $\eta =\left[g,\;h\right]$.*


These set of Tukey transformations modify the tail behaviour and the skewness relative to the base distribution, in this case the GP model which has finite dimensional Gaussian distributions point-wise in space. As such, the warped GP model will be able to produce a class of processes which, point-wise in space, will have distributions that are less symmetric around the mode. In particular they can accommodate left or right skewness, depending on the sign of *g* in the warping transform G. Furthermore, they can be heavier tailed than a Gaussian probability distribution, via leptokurtic transform H, and the strength of kurtosis relative to the Gaussian at each spatial point is controlled by parameter *h*. The Tukey g-and-h transformation is known to encompass a large range of distributions and this is further explained in [31,32]. This is one of the most desirable properties of this transformation.

Before proceeding in using this family of warping functions to define a warped GP spatial process, it will be useful to provide a few basic properties of this class of transform, that whilst are simple to mathematically verify, are also highly useful in developing further the specification of the properties of the resulting warped processes.

When working with warped GPs it is valuable to be able to evaluate the transformed finite dimensional density at a single location in space point-wise when performing tasks such as hyper-parameter estimation via the likelihood and also for spatial field reconstruction. Therefore, it is of relevance to ask if the warping function is invertible, as such a property leads one to obtaining a tractable expression for the finite dimensional distributions. We show in Proposition 1 the conditions under which the inverse of the warping function exists. Consequently, we can be sure that should we satisfy these conditions for the warping function, then in the case of the Tukey g-and-h transformation, it does accommodate a closed form distribution representation which will be tractable up to performing a simple univariate root search, as we will see later in Proposition 5.

**Proposition** **1**(Inverse of the Tukey g-and-h transformation)**.**
*The inverse of the Tukey g-and-h transformation $T_{gh}\left(W;\zeta_{\mathrm{gh}}\right)$ exists if $g\in R-\left\{0\right\}$ and $h\in R^{+}$*


**Proof.** See Appendix A. □

**Remark** **1.**
*The result in Proposition 1 affirms the existence of the inverse of this transformation but does not provide an analytical expression for the inverse. In practice, one needs to find the inverse using simple univariate root search computational methods in order to facilitate evaluation of the warped distributions or densities point-wise. The existence of the inverse is still critical to verify for the warping transform of the GP, as it will be used to facilitate expression of the spatial processes finite dimensional distributions, which will in turn be used for specification of the likelihood for use in calibration and spatial field estimation and reconstruction.*


It will also be useful to provide simple characterisations of the derivatives of the warping functions up to third order, as these will be used numerous times in the warped GP spatial field reconstruction estimation frameworks to capture the moments, mode and curvature approximations (covariance) point-wise. The derivatives of the Tukey g-and-h transform are given in Lemma 1 by simple algebra.

**Lemma** **1**(Derivatives of Tukey g-and-h transform)**.**
*The first three derivatives of the Tukey g-h transform are given by:*


(5)
$$\begin{array}{cc} T_{gh}^{\prime }\left(W;\zeta_{\mathrm{gh}}\right) & =hWT_{gh}\left(W;\zeta_{\mathrm{gh}}\right)+\exp\left(gW+\frac{hW^{2}}{2}\right), \end{array}$$



(6)
$$\begin{array}{cc} T_{gh}^{\prime \prime }\left(W;\zeta_{\mathrm{gh}}\right) & =hT_{gh}\left(W;\zeta_{\mathrm{gh}}\right)+hWT_{gh}^{\prime }\left(W;\zeta_{\mathrm{gh}}\right)+\exp\left(gW+\frac{hW^{2}}{2}\right)\left(hW+g\right), \end{array}$$



(7)
$$\begin{array}{cc} T_{gh}^{\prime \prime \prime }\left(W;\zeta_{\mathrm{gh}}\right) & =hWT_{gh}^{\prime \prime }\left(W;\zeta_{\mathrm{gh}}\right)+2hT_{gh}^{\prime }\left(W;\zeta_{\mathrm{gh}}\right)+\exp\left(gW+\frac{hW^{2}}{2}\right)\left({\left(hW+g\right)}^{2}+h\right). \end{array}$$




### Tukey Warped Spatial Gaussian Process

One may now proceed to developing a characterisation of a warped GP spatial process based around the Tukey family of skewness and kurtosis transformations. We begin by providing a definition of the GP that will set the notation we will use to specify the spatial trend and spatial covariance operator for the input GP spatial model. This GP model will serve as our base distribution (process) to generate non-Gaussian fields, i.e., warped GP processes. Then we will show how one can construct the warped GP model based on the Tukey g-and-h random field transform.

**Definition** **3.**
*Gaussian Process (GP) [14]:*

*Denote by $W\left(x\right):X\mapsto R$ a stochastic process parameterised by $\left\{x\right\}\in X$, where $X\subseteq R^{d}.$ Then, the random function $W\left(x\right)$ is a Gaussian process if all its finite dimensional distributions are Gaussian, where for any m∈N, the random vector $\left(W\left(x_{1}\right),W\left(x_{2}\right),\ldots ,W\left(x_{m}\right)\right)$ is jointly normally distributed [14].*


We can therefore interpret a GP as formally defined by the following class of random functions:$$\begin{array}{c} \begin{array}{cc} W & :=W\left(\cdot \right):X\mapsto R\\ \; & \left.s.t.\;W\left(\cdot \right)∼\mathrm{GP}\left(\mu \left(\cdot ;\theta \right),C\left(\cdot ,\cdot ;\Psi \right)\right),\;\mathrm{with}\;\right.\\ \mu \left(\cdot ;\theta \right) & :=E\left[W\left(\cdot \right)\right]:X\mapsto R,\\ C\left(\cdot ,\cdot ;\Psi \right) & :=\left.E\left[\left(W\left(\cdot \right)-\mu \left(\cdot ;\theta \right)\right)\left(W\left(\cdot \right)-\mu \left(\cdot ;\theta \right)\right)\right]:X\times X\mapsto R^{+}\right\}. \end{array} \end{array}$$
At each point the function valued process is characterised by the mean function μ(·;θ), parameterised by θ, and the spatial dependence between any two points is given by the covariance function (Mercer kernel) $C\left(\cdot ,\cdot ;\Psi \right)$, parameterised by Ψ, see detailed discussion in [14].

It will be useful to make the following notational definitions for the prior mean vector, cross correlation vector and auto-correlation matrix, respectively:$$\begin{array}{c} \begin{array}{cc} \left[\mu \left(x_{1};\theta \right),\mu \left(x_{2};\theta \right),\ldots ,\mu \left(x_{m};\theta \right)\right] & =\left[a_{1},a_{2},\ldots ,a_{m}\right]\\ k\left(x_{*},x_{1:m\;}\right) & :=\left[C\left(x_{*},x_{1}\right),C\left(x_{*},x_{2}\right),\ldots ,C\left(x_{*},x_{m}\right)\right]\in R^{1\times m}\\ K\left(x_{1:m\;},x_{1:m\;}\right) & :=\left[\begin{array}{ccc} C\left(x_{1},x_{1}\right) & \cdots & C\left(x_{1},x_{m}\right)\\ \vdots & \ddots & \vdots \\ C\left(x_{m},x_{1}\right) & \cdots & C\left(x_{m},x_{m}\right) \end{array}\right]\in X^{+}\left(R^{m}\right), \end{array} \end{array}$$
with $X^{+}\left(R^{m}\right)$ is the manifold of symmetric positive definite matrices. $C\left(x_{i},x_{j}\right)$ is the covariance function.

We now provide a formal definition of the Tukey g-and-h process.

**Definition** **4.**
*Spatial Tukey Warped Gaussian Process:*

*Denote by $Y\left(x\right):X\mapsto R$ a warped spatial stochastic process parameterised by points in space $\left\{x\right\}\in X\subseteq R^{2}$. Then, $Y\left(x\right)$ is a Tukey warped Gaussian Process if all its m-point finite dimensional distributions, for any m∈N, are jointly distributed according to the random vector obtained by an elementwise Tukey transformation of the underlying spatial Gaussian process on points $\left\{x_{1},\ldots ,x_{m}\right\}$ with $x_{i}\in X$, given by $\left(Y\left(x_{1}\right),\ldots ,Y\left(x_{m}\right)\right):=\left(T\left(W\left(x_{1}\right);\zeta \right),\ldots ,T\left(W\left(x_{m}\right);\zeta \right)\right)$ for any m∈N, where $\left(W\left(x_{1}\right),\ldots ,W\left(x_{m}\right)\right)$ are jointly normally distributed from a spatial Gaussian Process W.*


## 3. Properties of the Spatial Tukey Warped Gaussian Process

Having developed a basic definition of the warped GP model based on a Tukey family of warping functions, we can now study the basic properties of this spatial process by first exploring its finite dimensional distributions. Such basic finite dimensional multi-point characterisations are useful for preparing spatial field reconstruction estimators for these warped GP models. This starts with the model for a single point in space (univariate marginal warped GP model in Section 3.1), followed by specification of the properties of multiple-points in space (multivariate finite dimensional marginal warped GP model in Section 3.3).

### 3.1. Spatial Tukey Warped Gaussian Characterisation at a Single Point in Space

At any point in space $x\in X\subseteq R^{2}$ one may consider the properties of the single location warped GP model and its properties. In this case, at one location we may consider as input a standard GP with zero spatial mean and spatial covariance function *k* giving a Gaussian random variable W(x)∼N(μ,C(x,x)). Then the baseline reference Tukey g-and-h warped random variable is obtained by setting $\zeta_{\mathrm{gh}}:=\left[m_{0},m_{1},\mu ,C,g,h,\theta =1\right]$ where $g\in R-\left\{0\right\}$ and $h\in R^{+}$ to give:(8)$$\begin{array}{c} \begin{array}{cc} Y\left(x\right):=T_{gh}\left(W\left(x\right);\zeta_{\mathrm{gh}}\right) & =m_{0}+m_{1}\frac{\exp\left(gW(x)\right)-1}{g}\exp\left(\frac{hW{\left(x\right)}^{2}}{2}\right). \end{array} \end{array}$$
In the next proposition, we demonstrate that the warped Gaussian model will include the classical Gaussian model as a limiting case. This result is most readily proven for the setting of a single point in space, where we show that Y(x) has a Gaussian law under some special specification of the warping function or its parameters.

**Proposition** **2**(Limiting case: Gaussian distribution as a special case)**.**
*The random variable $Y\left(x\right):=T_{gh}\left(W\left(x\right);\zeta_{\mathrm{gh}}\right)$, where $\zeta_{\mathrm{gh}}:=\left[m_{0},m_{1},\mu ,C,g,h,\theta =1\right]$, in the limit g→0 and h→0, reduces to*

(9)
$$\begin{array}{c} Y\left(x\right)∼N\left(m_{0}+\mu ,m_{1}^{2}C\left(x,x\right)\right). \end{array}$$

*By setting $m_{0}=0,m_{1}=1$ we recover $Y\left(x\right)∼N\left(\mu ,C(x,x)\right)$, i.e., $Y\left(x\right)\overset{d}{=}W\left(x\right)$.*


**Proof.** See Appendix B. □

**Remark** **2.**
*The parameter g∈R modifies the skewness relative the base distribution, in this case the Gaussian. A positive value of g indicates a positive skew in the transformed distribution and a negative value of g indicates a negative skew. The parameter h indicates a transformation in the kurtosis relative to the base distribution and to guarantee existence of moments, one may consider a restricted range such as 0≤h≤1.*


We can also easily characterise the tail behaviour of the propose g-and-h warping transform to demonstrate how the *h* parameter in the H transform is able to produce leptokurtic behaviour in the resulting warped Gaussian density. One way to understand this is to demonstrate that the resulting warped distribution function has a property of regular variation, in other words it can be expressed as a power law tail decay and a slowly varying function.

Therefore, we briefly study regular variation of a distribution of the Tukey g-and-h distribution, which is a measure of the heaviness of tails of a distribution. Later, we connect this to the existence of the *n*-th moment. The index of the regular variation for the Tukey g-and-h distribution has been studied in [32,39], we summarise this result below for a generic base Gaussian distribution with $W∼N\left(\mu ,C(x,x)\right)$. It is important to understand under what conditions one has finite first and second order moments for a model with a specific kurtosis transform if one is to develop classes of linear spatial field reconstruction based on first and second moment local approximations. An advantage of the Tukey g-and-h warping function is that one can set conditions of the warping functions through simple parameter space restrictions to ensure existence of appropriate moments for use in spatial field reconstructions. The following result in Lemma 2 is meaningful to accommodate this analysis.

**Lemma** **2.**
*Consider a location x∈X and input GP random variable $W\left(x\right)∼N\left(\mu ,C(x,x)\right)$ then the Tukey g-and-h random variable $Y\left(x\right):=T_{gh}\left(W\left(x\right);\zeta_{\mathrm{gh}}\right)$ with parameters $\zeta_{\mathrm{gh}}:=\left[m_{0}=0,m_{1}=1,\mu ,C\left(x,x\right),g,h,\theta =1\right]$ has an index of regular variation for its survival function ${\bar{F}}_{Y(x)}\left(y\right):=1-F_{Y(x)}\left(y\right)$ given by:*

(10)
$$\begin{array}{cc} \lambda & =\frac{1}{h\;C(x,x)}. \end{array}$$



**Proof.** See Appendix C. □

### 3.2. Spatial Tukey Warped Gaussian Characterisation at Multiple Points in Space

One then readily extends the model development from the univariate Tukey g-and-h random variable, that characterises the warped GP model at a single location x∈X, to a multiple-location specification at points $x_{1},\ldots ,x_{m}\in X$ which requires specification of a multivariate Tukey g-and-h random vector $Y=\left[Y\left(x_{1}\right),\ldots ,Y\left(x_{m}\right)\right]$. Such extensions have for instance been specified in constructions in [4,31,35] where the single point to multiple point extension is obtained by applying an element-wise Tukey g-and-h transformation to a Multivariate Normal distribution, $W_{1:m}∼\mathrm{MVN}\left(w;\mu ,\Sigma \right)$ with mean vector μ and covariance matrix Σ such that $\Sigma_{ij}=C\left(x_{i},x_{j}\right)$ based on the GP covariance kernel *k* at two points in space $x_{i},x_{j}$. This gives for the *j*-th location $x_{j}$ the transformed random variable
(11)$$\begin{array}{c} Y_{j}:=T_{gh}\left(W_{j};\zeta_{gh}\right),\;\;j\in \left\{1,\cdots ,m\right\}. \end{array}$$
In understanding the basic properties of the resulting multivariate warped random vector for the Tukey g-and-h model at multiple spatial locations, we can study the exponential moments (Lemma 3) which will be instrumental in deriving many of the properties of the warped GP models spatial field reconstruction estimators.

Although the Tukey g-and-h multivariate random vector and the Tukey g-and-h processes have been defined previously in the aforementioned works. They have not been carefully studied from the perspective of deriving closed form results for their higher order cross moments or spatial moments of any order, including cross covariance, cross skewness and cross kurtosis. In this section, our goal is therefore to derive certain desirable properties which help to characterize the Tukey g-and-h processes further than just their definition.

To achieve these spatial cross moment characterisations it will be useful to first state a result for the quadratic exponential moments of a multivariate normal random vector. This form of moment structure will arise naturally when expressing the cross moments of the class of warped GP model developed under Tukey transforms. Furthermore, these closed form results will be useful for developing computationally efficient and interpretable spatial field reconstructions based on a spatial Tukey warped GP model class.

**Lemma** **3**(Exponential Moments of a Gaussian Random Vector)**.**
*Consider a m-variate random vector $W:={\left[W_{1},\ldots ,W_{m}\right]}^{T}$ distributed as $W∼\mathrm{MVN}\left(w;\mu ,\Sigma \right),$ with diagonal matrix $\Sigma_{v}$ such that ${\left[\Sigma_{v}\right]}_{i,i}=v_{i},i\in \left\{1,\ldots ,m\right\}$ and $u\in R^{m}$, then the quadratic exponential moments of W are given by:*

(12)
$$\begin{array}{cc} E\left[\exp\left(\frac{1}{2}W^{T}\Sigma_{v}W+u^{T}W\right)\right]\; & =\frac{{\left|\tilde{\Sigma }\right|}^{\frac{1}{2}}}{{\left|\Sigma \right|}^{\frac{1}{2}}}\exp\left(-\frac{1}{2}\left(\mu^{T}\Sigma^{-1}\mu -{\tilde{\mu }}^{T}{\tilde{\Sigma }}^{-1}\tilde{\mu }\right)\right), \end{array}$$

*where,*

(13)
$$\begin{array}{cc} \tilde{\Sigma } & ={\left(\Sigma^{-1}-\Sigma_{v}\right)}^{-1}, \end{array}$$


(14)
$$\begin{array}{cc} \tilde{\mu } & =\tilde{\Sigma }\left(u+\Sigma^{-1}\mu \right). \end{array}$$



**Proof.** See Appendix D. □

This result can now be developed into a core component, to characterise the raw cross moments and cross central moments, for the warped GP model. We again consider *m*-points in space in a vector $x_{1:m}:=\left[x_{1},\ldots ,x_{m}\right]$. Let us first describe the generic equation for an *m*-variate raw cross moment of a spatial Tukey warped GP model.

**Proposition** **3**(Spatial Tukey Warped GP Raw Moments for *m* Points in Space)**.**
*Consider the m-point spatial Tukey warped GP model with random vector $Y=\left[Y\left(x_{1}\right),\ldots ,Y\left(x_{m}\right)\right]$. Assume w.l.o.g. a standardised transform with location and scale parameters given by $m_{0}=0$, $m_{1}=1$ pointwise and $\zeta_{\mathrm{gh}}:=\left[0,1,g,h\right]$ selected such that existence of finite cross moments of orders $\left(n_{1},\ldots ,n_{m}\right)$ for $n_{i}\in N$ holds. Then the raw spatial m-point cross moments of orders $\left(n_{1},\ldots ,n_{m}\right)$ are given by:*

(15)
$$\begin{array}{c} \begin{array}{cc} {\overset{ˇ}{\mu }}_{gh}^{\left(n_{1},\ldots ,n_{m}\right)}: & =E\left[\prod_{j=1}^{m}{\left(Y\left(x_{j}\right)\right)}^{n_{j}}\right]\\ \; & =E\left[\prod_{j=1}^{m}{\left(T_{gh}\left(W\left(x_{j}\right);\zeta_{\mathrm{gh}}\right)\right)}^{n_{j}}\right]\\ \; & =\frac{1}{g^{\left(n_{1}+n_{2}+\ldots +n_{m}\right)}}\sum_{i_{1}=0}^{n_{1}}\ldots \sum_{i_{m}=0}^{n_{m}}{\left(-1\right)}^{\left(i_{1}+i_{2}+\ldots +i_{m}\right)}\frac{{\left|\tilde{\Sigma }\right|}^{\frac{1}{2}}}{{\left|\Sigma \right|}^{\frac{1}{2}}}\exp\left(-\frac{1}{2}\left(\mu^{T}\Sigma^{-1}\mu -{\left({\tilde{\mu }}^{\left(i\right)}\right)}^{T}{\tilde{\Sigma }}^{-1}{\tilde{\mu }}^{\left(i\right)}\right)\right), \end{array} \end{array}$$

*where*

(16)
$$\begin{array}{cc} \tilde{\Sigma } & ={\left(\Sigma^{-1}-\Sigma_{v}\right)}^{-1} \end{array}$$


(17)
$$\begin{array}{cc} {\tilde{\mu }}^{\left(i\right)} & =\tilde{\Sigma }\left(u^{\left(i\right)}+\Sigma^{-1}\mu \right), \end{array}$$


(18)
$$\begin{array}{cc} u^{\left(i\right)} & =g{\left[\left(n_{1}-i_{1}\right),\left(n_{2}-i_{2}\right),\ldots ,\left(n_{m}-i_{m}\right)\right]}^{T} \end{array}$$

*and $\Sigma_{v}$ is a diagonal matrix with known elements ${\left[\Sigma_{v}\right]}_{i,i}=hn_{i},i\in \left\{1,\ldots ,m\right\}.$*


**Proof.** See Appendix E. □

This general result is then instrumental in producing practical examples of raw *m*-location cross moments of the spatial Tukey warped GP model of orders $\left(n_{1},\ldots ,n_{m}\right)$ denoted by ${\overset{ˇ}{\mu }}_{gh}^{\left(n_{1},\ldots ,n_{m}\right)}$ as well as the centralised cross moments denoted by $\mu_{gh}^{\left(n_{1},\ldots ,n_{m}\right)}$, which are obtained by this same identity coupled with a binomial sum expansion to accommodate the centering transform at each of the locations $x_{1}$ to $x_{m}$.

This result generalises the previous results in the literature for moments of a Tukey g-and-h process which previously only explored pairwise (m=2 points) second and first order moments, where as this result obviously provides generalised m≥1 point cross moment expressions of any orders. This is a significant extension in characterising this family of warped GP models, which no-longer have sufficient statistics (functions) given solely by mean and covariance functions.

### 3.3. Practical cross Moment Special Cases for Spatial Field Reconstruction of Tukey Warped GP Models

The result in Proposition 3, whilst very general, can be used to readily express a few core identities of relevance to spatial field reconstruction estimators at multiple locations in space, when one assumes to have observations consistent with a class of spatial Tukey warped GP models.

In particular, we will state the moments and conditions for their existence in terms of the Tukey *h* parameter in the warping function H as well as the two-point cross moments.

**Corollary** **1**(Single Location Central Moments of Spatial Tukey Warped Gaussian Process)**.**
*The $n_{i}$-th central moment at location $x_{i}$ is given by:*

(19)
μgh(0,…,0,ni,0,…,0)=1gni1−nihCxi,xi∑i=0ni(−1)iniiexpnihai2+2aigni−i+ni−i2g2C(xi,xi)21−nihCxi,xi



As stated in Proposition 4 below, the results that allow one to connect the idea of regular variation to the existence of moments of a certain power are provided. One should then understand that as the tails of the distribution become heavier (as measured by the index of regular variation), the integrals involved in the calculation of moments may cease to converge to a finite value. As such, one can formally state the relationship between the moment existence and the regular variation index as follows in Proposition 4.

**Proposition** **4**(Existence of Moments of Single Location Spatial Tukey Warped GP)**.**
*For the $n_{i}$-th moment to exist in Corollary 1 we have to ensure that the following inequality holds:*

(20)
$$\begin{array}{cc} h & <\frac{1}{n_{i}C\left(x_{i},x_{i}\right)} \end{array}$$



**Proof.** The existence proof is based on the fact that the for the *n*-th moment to exist for a random variable, the index of regular variation of the random variable must be strictly greater than *n*. Thus, from Lemma 2, we get:
(21)$$\begin{array}{c} \begin{array}{cc} \lambda & >n,\\ \Rightarrow \frac{1}{h\sigma^{2}} & >n,\\ \Rightarrow h & <\frac{1}{n\sigma^{2}}. \end{array} \end{array}$$
where $\sigma^{2}=C\left(x_{i},x_{i}\right)$, for the Tukey g-and-h process at the location $x_{i}$. □

Next we can express, as a special case for two points $x_{i},x_{j}$ in space (m=2), the cross moments of order $\left(n_{i},n_{j}\right)$ as follows.

**Corollary** **2**(Two Location Cross Central Moments of Spatial Tukey Warped Gaussian Process)**.**
*The $n_{i}$-th and the $n_{j}$-th order cross moment between two locations $x_{i}$ and $x_{j}$*

(22)
μgh(0,…,0,ni,0,…,0,nj,0,…,0)=1gni+njDΣ∑ii=0ni∑ij=0nj(−1)ii+ijniiinjijexpCxj,xjμ˜1(i)2−|Σ|2DΣ2ai2|Σ|3DΣ×expCxi,xiμ˜2(i)2−|Σ|2DΣ2aj2−Cxi,xj+Cxj,xiμ˜1(i)μ˜2(i)−|Σ|2DΣ2aiaj|Σ|3DΣ,

*where:*

(23)
$$\begin{array}{c} \begin{array}{cc} \left|\Sigma \right| & =C\left(x_{i},x_{i}\right)C\left(x_{j},x_{j}\right)-C\left(x_{i},x_{j}\right)C\left(x_{j},x_{i}\right),\\ {\tilde{\mu }}_{1}^{\left(i\right)} & =\left|\Sigma \right|\left(a_{i}\left(1-hn_{j}C\left(x_{j},x_{j}\right)\right)+a_{j}hn_{j}C\left(x_{i},x_{j}\right)+g\left(\left(C\left(x_{i},x_{i}\right)-hn_{j}\left|\Sigma \right|\right)\left(n_{i}-i_{i}\right)+C\left(x_{i},x_{j}\right)\left(n_{j}-i_{j}\right)\right)\right),\\ {\tilde{\mu }}_{2}^{\left(i\right)} & =\left|\Sigma \right|\left(a_{j}\left(1-hn_{i}C\left(x_{i},x_{i}\right)\right)+a_{i}hn_{i}C\left(x_{j},x_{i}\right)+g\left(\left(C\left(x_{j},x_{j}\right)-hn_{i}\left|\Sigma \right|\right)\left(n_{j}-i_{j}\right)+C\left(x_{j},x_{i}\right)\left(n_{i}-i_{i}\right)\right)\right),\\ D_{\Sigma } & =h^{2}n_{i}n_{j}\left|\Sigma \right|-h\left(n_{i}C\left(x_{i},x_{i}\right)+n_{j}C\left(x_{j},x_{j}\right)\right)+1. \end{array} \end{array}$$



**Proof.** This is direct application for points $x_{i},x_{j}$ of Proposition 3. □

We note that with these results one may simply recover the mean at location $x_{i}$ with $n_{i}=1$ giving: (24)$$\begin{array}{cc} E\left[Y\left(x_{i}\right)\right]=\mu_{gh}^{\left(0,\ldots ,0,1,0,\ldots ,0\right)} & =\frac{1}{g\sqrt{1-hC\left(x_{i},x_{i}\right)}}\left[\exp\left(\frac{ha_{i}^{2}+g^{2}C\left(x_{i},x_{i}\right)+2a_{i}g}{2(1-hC\left(x_{i},x_{i}\right))}\right)-\exp\left(\frac{ha_{i}^{2}}{2(1-hC\left(x_{i},x_{i}\right))}\right)\right]. \end{array}$$
and the variance at location $x_{i}$ with $n_{i}=2$ as follows: (25)$$\begin{array}{c} \begin{array}{cc} Var\left[Y\left(x_{i}\right)\right]=\kappa_{gh}^{\left(0,\ldots ,0,2,0\ldots ,0\right)} & =\frac{1}{g^{2}\sqrt{1-2hC\left(x_{i},x_{i}\right)}}(\exp\left(\frac{ha_{i}^{2}+2a_{i}g+2g^{2}C\left(x_{i},x_{i}\right)}{1-2hC\left(x_{i},x_{i}\right)}\right)\\ \; & -2\exp\left(\frac{2ha_{i}^{2}+2a_{i}g+g^{2}C\left(x_{i},x_{i}\right)}{2\left(1-2hC\left(x_{i},x_{i}\right)\right)}\right)+\exp\left(\frac{ha_{i}^{2}}{1-2hC\left(x_{i},x_{i}\right)}\right))\\ \; & -{\left(\mu_{gh}^{\left(0,\ldots ,0,1,0,\ldots ,0\right)}\right)}^{2}. \end{array} \end{array}$$

It will also be practically useful for the spatial field reconstructions to state the cross spatial covariance between the spatial field at new location $x_{*}$ and existing observed points $x_{1:N}$. These cross spatial covariance results will be critical for derivation of the multiple point Spatial-Best Linear Unbiased Estimator class we term the (S-BLUE) spatial field reconstruction. Consider the joint description of a location $x_{*}$ and the locations of the observations $Y_{1:N}$ by the set $\left\{x_{1},x_{2},\ldots ,x_{N},x_{*}\right\}$. We also define $\kappa_{gh}^{\left(0,\ldots ,0,1,0,\ldots ,0,1,0,\ldots ,0\right)}=\mu_{gh}^{\left(0,\ldots ,0,1,0,\ldots ,0,1,0,\ldots ,0\right)}-\mu_{gh}^{\left(0,\ldots ,0,1,0,\ldots ,0,0,0,\ldots ,0\right)}\mu_{gh}^{\left(0,\ldots ,0,0,0,\ldots ,0,1,0,\ldots ,0\right)}$ as the covariance at any two locations $x_{i}$ and $x_{j}$. Then we obtain the following quantities using the aforementioned derivations applied to obtain these cross moments:$$\begin{array}{c} \begin{array}{cc} \mu_{p_{*}} & =a_{*}+k\left(x_{*},x_{1:N\;}\right)K^{-1}\left(x_{1:N\;},x_{1:N\;}\right)\left(T_{gh}^{-1}\left(Y_{1:N}-E\left[Y_{1:N}\right]\right)\right)\\ Cov\left(Y_{*},Y_{1:N}\right) & :=\left[\kappa_{gh}^{\left(1,0,\ldots ,0,1\right)},\kappa_{gh}^{\left(0,1,\ldots ,0,1\right)},\ldots ,\kappa_{gh}^{\left(0,0,\ldots ,1,1\right)}\right]\in R^{1\times N}\\ Cov\left(Y_{1:N},Y_{1:N}\right) & :=\left[\begin{array}{ccc} \kappa_{gh}^{\left(1,0,\ldots ,0,0\right)} & \cdots & \kappa_{gh}^{\left(1,0,\ldots ,1,0\right)}\\ \vdots & \ddots & \vdots \\ \kappa_{gh}^{\left(1,0,\ldots ,1,0\right)} & \cdots & \kappa_{gh}^{\left(0,0,\ldots ,1,0\right)} \end{array}\right]\in X^{+}\left(R^{N}\right), \end{array} \end{array}$$

Having derived these properties of the Tukey g-and-h random field we are now ready to derive the results for various field reconstruction metrics.

## 4. Field Reconstruction for Spatial Tukey Warped Gaussian Processes

In many real world settings, for spatial field modelling and analysis, the spatial fields finite dimensional distributions or process may not be used explicitly in modelling and instead a selection of relevant point estimators will be used to characterise a statistical summary of the warped Gaussian process, in our case the Tukey g-and-h warped spatial processes. In this section, we propose estimators and show how to solve, for a variety of practically relevant spatial field point estimators, often called in spatial statistics “spatial field reconstruction” or “spatial field approximation/estimation” [7,8]. In our case, we tailored the spatial field reconstructions to three classes of estimator with varying efficacy and computational cost in construction.

We have formulated this spatial field reconstruction context in terms of classical risk minimization through a loss function formulating the class of estimator which is resolved with respect to the spatial process, namely the warped GP model. We are assuming the warped GP model is calibrated and here the exercise is not in calibration of this models hyper parameters for the spatial covariance kernel and warping skew and kurtosis parameters, namely *g* and *h*, respectively, but rather in regards to approximating efficiently point estimators for the spatial field at any collection of target locations.

In this regard we derive five different estimators for various important quantities often considered in spatial field reconstruction problems. The multi-point Minimum Mean Squared Error (MMSE) estimators, the multiple point Maximum A-Posteriori (MAP) estimators, an efficient class of multiple-point linear estimators based on the Spatial-Best Linear Unbiased (S-BLUE) estimators, and two multi-point threshold exceedance based estimators, namely the Spatial Regional and Level Exceedance estimators.

Throughout the remainder of the manuscript, in order to simplify the notation, we will denote process $Y(x_{i})$ at location $x_{i}$ simply by $Y_{i}$. To perform spatial field reconstruction, under the assumption that the data generating process for the spatial observations is a spatial Tukey warped GP, one may approach the problem under a Bayesian paradigm. One may then develop field reconstruction estimators, at an arbitrary location $x_{*}$, at which one has not observed the process, based on the predictive posterior density at any location in space, $x_{*}\in X$, denoted $p\left(Y_{*}|Y_{1:N}\right)$. Here we denote $Y_{1:N}$ as the measurements obtained from the sensors that observe the process at *N* locations $x_{1:N}\in X$. This approach is widely used and is the standard in many applications in sensor networks [1,2,3,4,6,7,8,9,10]

To proceed with development of the conditional probability of prediction of the distribution of the spatial field, at unobserved location $x_{*}$, we need to specify the warped spatial GP’s density, given in Lemma 4. This result shows how to predict the spatial field at the new unobserved location $x_{*}$ conditional on the observations of the warped GP spatial field $Y_{1:N}$ at observed locations $x_{1:N}$. Due to the structure of the warping spatial process developed, we are able to analytically express the first two moments for the predictive condition mean and covariance in close form. This is critical for efficient development of point estimators for the spatial field reconstructions that will be developed in following sections.

**Lemma** **4**(Predictive Posterior Density at Location $x_{*}$)**.**
*The predictive posterior density $p\left(Y_{*}|Y_{1:N},x_{1:N\;},x_{*}\right)$ is given by:*

(26)
$$\begin{array}{c} \begin{array}{c} Y_{*}|Y_{1:N},x_{1:N\;},x_{*}∼\mathrm{GH}\left(y_{*};{ı}_{gh}\right), \end{array} \end{array}$$

*where ${ı}_{gh}:=\left[m_{0}=0,m_{1}=1,\mu_{p_{*}},\sigma_{p_{*}}^{2},g,h,\theta =1\right]$ and*

(27)
$$\begin{array}{c} \begin{array}{cc} \mu_{p_{*}} & =a_{*}+k\left(x_{*},x_{1:N\;}\right)K^{-1}\left(x_{1:N\;},x_{1:N\;}\right)\left(T_{gh}^{-1}\left(Y_{1:N}-E\left[Y_{1:N}\right]\right)\right),\\ \sigma_{p_{*}}^{2} & =k\left(x_{*},x_{*}\right)-k\left(x_{*},x_{1:N\;}\right)K^{-1}\left(x_{1:N\;},x_{1:N\;}\right)k\left(x_{1:N\;},x_{*}\right). \end{array} \end{array}$$

*Note that $E\left[Y_{1:N}\right]$ here denotes the prior mean of all the observations $Y_{1:N}$ and $T_{gh}^{-1}\left(.\right)$ denotes a pointwise application of the inverse.*


**Proof.** A derivation of this conditional predictive density for a general g-and-h density with Guassian input process is given in [31]. □

Furthermore, we may derive based on this predictive density the first two predictive conditional moments which we state in the following corollaries describing the posterior mean and posterior variance of the Tukey g-h process at some arbitrary location $x_{*}$. There are obtained explicitly using results derived in Proposition 3 and the distribution characteristics specified in Section 3.

**Corollary** **3**(Posterior Predictive Mean at an Arbitrary Location $x_{*}$)**.**
(28)$$\begin{array}{cc} E\left[Y\left(x_{*}\right)|Y_{1:N},x_{1:N\;},x_{*}\right] & =\frac{1}{g\sqrt{1-h\sigma_{p_{*}}^{2}}}\left[\exp\left(\frac{h\mu_{p_{*}}^{2}+g^{2}\sigma_{p_{*}}^{2}+2\mu_{p_{*}}g}{2(1-h\sigma_{p_{*}}^{2})}\right)-\exp\left(\frac{h\mu_{p_{*}}^{2}}{2(1-h\sigma_{p_{*}}^{2})}\right)\right]. \end{array}$$

**Corollary** **4**(Posterior Predictive Variance at an Arbitrary Location $x_{*}$)**.**
(29)$$\begin{array}{c} \begin{array}{cc} Var\left[Y\left(x_{*}\right)|Y_{1:N},x_{1:N\;},x_{*}\right] & =\frac{1}{g^{2}\sqrt{1-2h\sigma_{p_{*}}^{2}}}(\exp\left(\frac{h\mu_{p_{*}}^{2}+2\mu_{p_{*}}g+2g^{2}\sigma_{p_{*}}^{2}}{1-2h\sigma_{p_{*}}^{2}}\right)\\ \; & -2\exp\left(\frac{2h\mu_{p_{*}}^{2}+2\mu_{p_{*}}g+g^{2}\sigma_{p_{*}}^{2}}{2\left(1-2h\sigma_{p_{*}}^{2}\right)}\right)+\exp\left(\frac{h\mu_{p_{*}}^{2}}{1-2h\sigma_{p_{*}}^{2}}\right))\\ \; & -{\left(E\left[Y\left(x_{*}\right)|Y_{1:N},x_{1:N\;},x_{*}\right]\right)}^{2}. \end{array} \end{array}$$

### 4.1. Spatial Field Reconstruction Risk Functionals and Loss Functions

In developing estimators formally for the spatial field reconstruction we introduce a set of different loss functions. These will characterise the risk functionals to be minimized with respect to the warped GP model in order to represent the spatial field reconstructions of different estimator types. We have developed five different spatial field reconstruction estimators that we formalise below based on the conditional predictive posterior distribution.

Based on the predictive posterior density, various point estimators, like the MMSE and the MAP estimators can be derived. These estimators provide a pointwise estimator of the intensity of the spatial filed, ${\hat{Y}}_{*}$ at location $x_{*}$. This enables us to reconstruct the whole spatial field by evaluating ${\hat{Y}}_{*}$ on a fine grid of points.

We first define a set of objective functions that covers different aspects of spatial regression and hence we calculate different estimators, each one having its special characteristic. We then derive explicitly each of the estimators in closed form expressions.

(1)Minimum Mean Squared Estimator (MMSE) estimatorThe MMSE estimator minimises the MSE by utilising a squared loss function:
(30)$$\begin{array}{c} \begin{array}{c} L\left(Y_{*},{\hat{Y}}_{*}\right):={\left(Y_{*}-{\hat{Y}}_{*}\right)}^{2}. \end{array} \end{array}$$Then, the MMSE spatial field reconstruction estimator is given by
(31)$$\begin{array}{c} \begin{array}{cc} {\hat{Y}}_{*}^{\mathrm{MMSE}} & =\underset{y}{arg min}E\left[L\left(Y_{*},{\hat{Y}}_{*}\right)|Y_{1:N},x_{1:N\;},x_{*}\right]\\ \; & =\underset{y}{arg min}E\left[{\left(Y_{*}-{\hat{Y}}_{*}\right)}^{2}|Y_{1:N},x_{1:N\;},x_{*}\right]\\ \; & =E\left[Y_{*}|Y_{1:N},x_{1:N\;},x_{*}\right]\\ \; & =\int_{-\infty }^{\infty }Y_{*}p\left(Y_{*}|Y_{1:N},x_{1:N\;},x_{*}\right)d\;Y_{*}. \end{array} \end{array}$$The conditional expectation is the estimate that minimises the conditional MMSE and the minimum value is given by the conditional variance shown below:
(32)$$\begin{array}{c} \sigma_{Y*|Y_{1:N}}^{2}=E\left[{\left(Y_{*}-E\left[Y_{*}|Y_{1:N},x_{1:N\;},x_{*}\right]\right)}^{2}|Y_{1:N},x_{1:N\;},x_{*}\right]. \end{array}$$(2)Maximum A-Posteriori (MAP) estimator:The MAP estimator is the mode of the predictive posterior density and utilizes the *0/1 loss function*:
(33)$$\begin{array}{c} \begin{array}{c} L\left(Y_{*},{\hat{Y}}_{*}\right):=\left\{\begin{array}{cc} 0 & ,\mathrm{If}\;\;\;\;Y_{*}={\hat{Y}}_{*}\\ 1 & ,\mathrm{Otherwise} \end{array}\right. \end{array} \end{array}$$Then, the MAP spatial field reconstruction estimator is given by
(34)$$\begin{array}{c} \begin{array}{cc} {\hat{Y}}_{*}^{\mathrm{MAP}} & =\underset{y^{*}}{arg max}p\left(Y_{*}|Y_{1:N},x_{1:N\;},x_{*}\right). \end{array} \end{array}$$(3)Spatial-Best Linear Unbiased Estimator (S-BLUE):The S-BLUE utilises the same loss function as the MMSE estimator, but is restricted to the linear family of estimators, given by:
(35)$$\begin{array}{c} \begin{array}{cc} {\hat{Y}}_{*}^{S\text{-}\mathrm{BLUE}} & =\underset{a_{*},B_{*}}{arg min}E\left[{\left(Y_{*}-a_{*}+B_{*}^{T}Y_{1:N}\right)}^{2}|Y_{1:N},x_{1:N\;},x_{*}\right], \end{array} \end{array}$$
where $a_{*}\in R$ and $B_{*}^{T}\in R^{1\times N}$(4)Spatial Regional and Level Exceedance estimators:There are a few ways of characterizing the spatial exceedance of these processes, either through the region or the location of the spatial exceedance given a user defined threshold and the tolerance. The other characteristic is the level at which the random process exceeds the given threshold at a given location. We define these characterizations mathematically as follows:(a)Spatial-Regional Right Tail Exceedance (S-RTE):
(36)$$\begin{array}{c} D_{x}\left(x_{*};Y_{1:N},x_{1:N\;},T,\alpha \right)=\left\{x_{*}:P\left(Y_{*}>T|Y_{1:N},x_{1:N\;},x_{*}\right)\geq 1-\alpha \right\}, \end{array}$$(b)Spatial-Regional Left Tail Exceedance (S-LTE):
(37)$$\begin{array}{c} D_{x}\left(x_{*};Y_{1:N},x_{1:N\;},T,\alpha \right)=\left\{x_{*}:P\left(Y_{*}<T|Y_{1:N},x_{1:N\;},x_{*}\right)\geq 1-\alpha \right\}, \end{array}$$
where *T* is the pre-defined threshold and α is the user defined tolerance. Here we are interested in the locations at which the above exceedance is observed. The function $D_{x}\left(x_{*};Y_{1:N},x_{1:N\;},T,\alpha \right):X\mapsto \left\{0,1\right\}$, i.e., it is a binary map in space indicating the locations $x_{*}$ at which the process exceeds a pre-defined threshold with a specified probability α. These locations may form points, lines or surfaces in $X\subseteq R^{2}$. It is useful to define exceedances on both tails due to the presence of skewness and kurtosis which affects the tail behaviour asymmetrically.(c)Spatial-Level Exceedance (S-LE):
(38)$$\begin{array}{c} D_{\alpha }=\inf\left\{\alpha :P\left(Y_{*}>T|Y_{1:N},x_{1:N\;},x_{*}\right)\geq 1-\alpha \right\}. \end{array}$$Here we are interested to find the minimum quantile level α at which the process exceeds the given threshold *T* at a given location $x_{*}$. The function $D_{\alpha }\left(x_{*};Y_{1:N},x_{1:N\;},T\right):X\mapsto \left(0,1\right)$, i.e.,

Now that we have specified various metrics by which the field reconstruction can be carried out, we detail each of the estimators.

### 4.2. Spatial Field Reconstruction Estimator Derivations

All the aforementioned objective functions except for the S-BLUE require the knowledge of the predictive posterior distribution conditioned on the observations and the sensor locations, given in Lemma 4. We begin by deriving the classical predictive spatial field estimator based on the minimum mean square error for the warped spatial GP.

**Lemma** **5**(MMSE spatial field reconstruction estimator [31])**.**
*The MMSE spatial field reconstruction function is given by the mean of the predictive posterior which is of the form similar to the one described in Corollary 3 and is given by:*

(39)
$$\begin{array}{c} \begin{array}{cc} {\hat{Y}}_{*|Y_{1:N}}^{\mathrm{MMSE}} & =\frac{1}{g\sqrt{1-h\sigma_{p_{*}}^{2}}}\left[\exp\left(\frac{h\mu_{p_{*}}^{2}+g^{2}\sigma_{p_{*}}^{2}+2\mu_{p_{*}}g}{2(1-h\sigma_{p_{*}}^{2})}\right)-\exp\left(\frac{h\mu_{p_{*}}^{2}}{2(1-h\sigma_{p_{*}}^{2})}\right)\right]. \end{array} \end{array}$$

*The Mean Squared Error (MSE) is given by the variance of the predictive posterior which is of the form similar to the one described in Corollary 4 and is given by:*

(40)
$$\begin{array}{c} \begin{array}{c} \mathrm{SMSE}\left(x_{*}\right)=\sigma_{Y*|Y_{1:N}}^{2}=\frac{1}{g^{2}\sqrt{1-2h\sigma_{p_{*}}^{2}}}[\exp\left(\frac{h\mu_{p_{*}}^{2}+2\mu_{p_{*}}g+2g^{2}\sigma_{p_{*}}^{2}}{1-2h\sigma_{p_{*}}^{2}}\right)\\ -2\exp\left(\frac{2h\mu_{p_{*}}^{2}+2\mu_{p_{*}}g+g^{2}\sigma_{p_{*}}^{2}}{2\left(1-2h\sigma_{p_{*}}^{2}\right)}\right)+\exp\left(\frac{h\mu_{p_{*}}^{2}}{1-2h\sigma_{p_{*}}^{2}}\right)]-{\left({\hat{Y}}_{*}^{\mathrm{MMSE}}\right)}^{2}. \end{array} \end{array}$$



**Remark** **3.**
*The expression for the uncertainty associated with the MMSE estimate is shown in Equation (Equation 40). We can see that the expression depends on the predictive posterior density parameter $\mu_{p_{*}}$, which in turn depends on the observations $Y_{1:N}$ as seen from Lemma 4. This is significantly different from the GP, where the uncertainty does no depend on the observations and only depends on the covariances between the locations at which the data were observed. This influences applications that involve experimental design problems like sensor placement, sensor selection, etc, in that just knowing the information of the locations alone is no longer sufficient for the Tukey g-h process and repeated experiments may need to be performed for evaluating the characteristics of just one location set.*


**Remark** **4.**
*Note that the existence of the MMSE at a location $x_{*}$ is given by, $h<\frac{1}{n\sigma_{p_{*}}^{2}}$. We also know that $\sigma_{p_{*}}^{2}\leq k\left(x_{*},x_{*}\right)$. This means that for a given value of the parameter h, there may arise a case when the Tukey g-and-h population mean at a location does not exist but the conditional mean can exist. This is because as we gather observations around the location of interest, we reduce $\sigma_{p_{*}}^{2}$ and hence the effect of the kurtosis parameter on the conditional MMSE.*


**Proposition** **5**(MAP spatial field reconstruction estimator:)**.**
*The spatial MAP estimator is obtained from the solution to the following optimisation problem:*

(41)
$$\begin{array}{cc} {\hat{Y}}_{*}^{\mathrm{MAP}} & =T_{gh}\left(w_{*}^{o};\zeta_{gh}\right) \end{array}$$

*where $w_{*}^{o}$ is the solution to the following optimsation problem.*

(42)
$$\begin{array}{cc} w_{*}^{o} & =\underset{w_{*}}{arg max}\frac{\exp\left(-\frac{{\left(w_{*}-\mu_{p_{*}}\right)}^{2}}{2\sigma_{p_{*}}^{2}}\right)}{hw_{*}\frac{\exp\left(gw_{*}\right)-1}{g}\exp\left(\frac{hw_{*}^{2}}{2}\right)+\exp\left(gw_{*}+\frac{hw_{*}^{2}}{2}\right)} \end{array}$$



**Proof.** See Appendix F. □

**Remark** **5.**
*The optimisation problem in specified in Proposition 5, to obtain the MAP estimate of the Tukey g-and-h process at a location $x_{*}$ does not admit analytical solutions and hence has to be dealt with numerically. To this end, one can solve this one-dimensional optimisation through standard techniques such as Newton methods adopting a simple gradient descent (since we can obtain the derivatives of this expression at all values of $w_{*}$).*


**Proposition** **6**(MAP spatial variance:)**.**
*The variance of the spatial MAP estimator is given by the hessian of the posterior GH distribution at a location $x_{*}$, evaluated at the optimum value $w_{*}^{o}$ obtained from Proposition 5. It is represented as follows:*

(43)
$$\begin{array}{c} \sigma_{{*}^{MAP}}^{2}=\frac{{\left(T_{gh}^{\prime }\left(w_{*}^{o}\right)\right)}^{2}f_{W}^{\prime \prime }\left(w_{*}^{o}\right)+2f_{W}^{\prime }\left(w_{*}^{o}\right)T_{gh}^{\prime \prime }\left(w_{*}^{o}\right)\left(T_{gh}^{\prime \prime }\left(w_{*}^{o}\right)-T_{gh}^{\prime }\left(w_{*}^{o}\right)\right)-f_{W}\left(w_{*}^{o}\right)T_{gh}^{\prime \prime \prime }\left(w_{*}^{o}\right)T_{gh}^{\prime }\left(w_{*}^{o}\right)}{{\left(T_{gh}^{\prime }\left(w_{*}^{o}\right)\right)}^{3}} \end{array}$$

*where $w_{*}^{o}$ is the mode of the GH distribution at $x_{*}$ and $f_{W}\left(.\right)$ is the Gaussian pdf with mean and variance as derived in Lemma 4.*


**Proof.** See Appendix G. □

We now state a few corollaries for finding the solution to the optimisation problem posed for finding the MAP estimate at any arbitrary location. These are the cases when the parameters g→0 and h→0.

**Proposition** **7**(S-BLUE spatial field reconstruction estimator:)**.**
*The Spatial-Best Linear Unbiased Estimator (S-BLUE) at any location $x_{*}$ is given by:*

(44)
$$\begin{array}{cc} {\hat{Y}}_{*}^{S\text{-}\mathrm{BLUE}} & ={\hat{a}}_{*}+{\hat{B}}_{*}^{T}\left(Y_{1:N}-E\left[Y_{1:N}\right]\right) \end{array}$$

*where ${\hat{a}}_{*}=\mu_{p_{*}}$ and $\hat{B}=Cov^{-1}\left(Y_{1:N},Y_{1:N}\right)Cov\left(Y_{*},Y_{1:N}\right)$. In addition, $E\left[Y_{1:N}\right]$ is just the prior mean of the observations $Y_{1:N}$.*


The exceedance problem of the Tukey g-and-h process, as described in the objective functions, given by Equations (Equation 36) and (Equation 37) are reduced to the problem of obtaining the multivariate quantile level sets of the process at *d* locations. First we look at univariate description of the quantile function at just one location. Then we will see that the existence of a tractable quantile Tukey g-and-h function enables us to reduce the exceedance problems to the description of these quantile function along with the user given parameters.

To proceed with these results it will be useful to state the following Lemma 6 that provides expressions for the marginal and conditional quantile functions of the Tukey g-and-h process.

**Lemma** **6**(Marginal and Conditional Quantile functions of the Tukey g-and-h process)**.**

*(1)* 
*Marginal Quantile Function:*

*The univariate marginal quantile function of the Tukey g-and-h process at any arbitrary location $x_{k}$ is described as follows:*

(45)
$$\begin{array}{c} Q_{Y_{k}}\left(\alpha \right)=T_{gh}\left(q_{k}\left(\alpha \right);{ı}_{gh}\right) \end{array}$$


*where $q_{k}\left(\alpha \right)=a_{k}+\sqrt{C\left(x_{k},x_{k}\right)}\Phi^{-1}\left(\alpha \right)$, and $\Phi^{-1}\left(\alpha \right)$ denotes the inverse cdf of the standard normal distribution.*
*(2)* 
*Conditional Quantile Function:*

*The univariate conditional quantile function of the Tukey g-and-h process at any arbitrary location $x_{*}$ is described as follows:*

(46)
$$\begin{array}{c} Q_{Y_{*}|Y_{1:N}}\left(\alpha \right)=T_{gh}\left(q_{*}\left(\alpha \right);{ı}_{gh}\right) \end{array}$$

*where $q_{*}\left(\alpha \right)=\mu_{p_{*}}+\sigma_{p_{*}}\Phi^{-1}\left(\alpha \right)$. Furthermore, $\Phi^{-1}\left(\alpha \right)$ is the inverse cdf of the standard normal distribution.*



**Proof.** See Appendix H. □

**Proposition** **8**(Spatial region of right tail exceedance estimator)**.**
*The region of right tail exceedance described in Equation (Equation 36) is reduced to finding the quantile level sets at the user given tolerance to see if it exceeds the given threshold. This is a binary estimate at location $x_{*}$, that is characterized by:*

(47)
$$\begin{array}{c} {\hat{Y}}_{*}^{S\text{-}\mathrm{RTE}}=\left\{\begin{array}{c} 1,\;\mathrm{if}\;q_{*}\left(\alpha \right)>T_{gh}^{-1}\left(T;{ı}_{gh}\right),\\ 0,\;\mathrm{Otherwise} \end{array}\right. \end{array}$$

*where $q_{*}\left(\alpha \right)=\mu_{p_{*}}+\sigma_{p_{*}}\Phi^{-1}\left(\alpha \right)$ is the quantile as seen from Lemma 6. The region is created by accumulating all the locations that produce 1 in the binary estimate.*


**Remark** **6.**
*We note that derivation of the left tail exceedance spatial field reconstruction estimator follows analogously the result obtained in Proposition 8 with a reversal of the inequality and a quantile level (1−α)∈[0,1].*


**Proposition** **9**(Spatial level of exceedance estimator)**.**
*The region of exceedance described in Equation (Equation 38) is reduced to finding the quantile level sets at the user given tolerance to see if it exceeds the given threshold. This is a binary estimate at location $x_{*}$, that is characterized by:*

(48)
$$\begin{array}{c} {\hat{Y}}_{*}^{S\text{-}\mathrm{LE}}=\inf\left\{\alpha :w_{*}\left(\alpha \right)>T_{gh}^{-1}\left(T;{ı}_{gh}\right)\right\} \end{array}$$

*where $w_{*}\left(\alpha \right)=\mu_{p_{*}}+\sigma_{p_{*}}\Phi^{-1}\left(\alpha \right))$ as seen in Lemma 6.*


Note that in Propositions 8 and 9, we used $x_{*}$ to denote the unobserved spatial field location. In these cases the quantile function is essentially the conditional quantile function from the conditional predictive posterior distribution obtained in Lemma 4.

## 5. Experimental Results: Warped Gaussian Process Spatial Field Reconstructions

In this section, we look at results concerning spatial field reconstruction based on a warped GP model of the Tukey g-and-h family of spatial processes. We consider both central estimators of spatial field as well as exceedance spatial maps. We compare the warped GP reconstructions of the Tukey g-and-h processes to those one would obtain had one miss-specified the model and applied a classical standard GP field reconstruction. It is of interest to assess the ability of the Tukey g-and-h process to detect attributes of spatial skewness and kurtosis, compared to the GP in a range of different settings. We consider both synthetic as well as real data studies to illustrate the Tukey g-and-h model developed.

### 5.1. Simulation Setup

In these case studies we model the physical phenomenon or spatial process that is partially observed by a Tukey transformation of an underlying GP over a spatial grid where $x\in \left[-5,5\right]$ and $y\in \left[-5,5\right]$. We work with the well known squared exponential kernel, with a known unit variance and known length-scale. We also subtract the mean to obtain a zero mean GP. Furthermore, the parameters g,h are assumed to be known and are set to different values within the allowable range, to test various cases. This allows us to demonstrate, on a synthetic case study, the importance of developing spatial field reconstructions that accommodate warped spatial GP structures for spatial skewness and kurtosis by comparing the loss in accuracy if one applies just a standard GP model.

Moreover, we have *N* sensors with known locations, spread uniformly over the spatial grid region of interest. We vary *N* to study the effect of the number of sensors on different regression estimators. To this end, we use the normalised spatial mean squared error (N-MSE) to characterize the performance of the different estimators and we define it as follows:$$\begin{array}{c} N\text{-}\mathrm{MSE}=\frac{1}{J\times T}\sum_{j=1}^{J}\sum_{i=1}^{T}\frac{{\left(\hat{Y}\left(x_{*}^{\left(ij\right)}\right)-Y\left(x^{\left(ij\right)}\right)\right)}^{2}}{maxY-minY} \end{array}$$
where *J* is the number of spatial locations to regress and *T* is the number of trials of the Tukey g-and-h process simulation. In addition, $maxY=\max_{i,j}x^{\left(ij\right)}$ is the maximum value of the simulated process across all locations and trials and similarly $minY=\min_{i,j}x^{\left(ij\right)}$ is the minimum value of the simulated process.

We outline the following test scenarios studied in this setting.

(1)Scenario 1: We compare the performance of the different estimators such as, MMSE (Equation (Equation 39)), S-MAP (Equation (Equation 41)), S-BLUE (Equation (Equation 35)) to the Gaussian Process estimate in various scenarios such as, small *g* and *h* with low spatial correlation, high spatial correlation and other such combinations of *g*, *h* and *l*.(2)Scenario 2: We characterize the exceedance estimation ability of the Tukey g-and-h process as shown in Proposition 8 (*g*,*h*), through spatial ROC curves compared to a Gaussian process.(3)Scenario 3: We adopt real data case studies for the US and we apply our estimators on the noisy observed US precipitation data sets [40]. We consider a spatial field reconstruction that involves latitudes between 34 and 43 degrees and longitudes between −110 and −100 degrees and we compare the different spatial field reconstruction estimators and their performance on real data versus a classical Gaussian process model.

Before we look at the simulation results, we show the actual differences in the distribution of the Tukey g-and-h finite dimensional distribution, with the parameters that are used in the simulation case studies, in order to compare to the Gaussian distribution, see Figure 1. This allows us to demonstrate clearly that at each spatial field location one would expect that skewness and kurtosis will be meaningful to consider in constructing the spatial field reconstruction estimators.

### 5.2. Scenario 1: Spatial Field Reconstruction on Synthetic Data

In this section, we present the estimation performance of the Spatial-BLUE estimator, GP estimator, Spatial-MAP estimator and the Spatial MMSE estimator. The results here are shown for a synthetic data set constructed with given parameter values for *g*, *h* and *l*, using a squared exponential kernel. The study utilised 50 realisations of the Tukey g-and-h random field and the average spatial Mean Squared Error reported below is the average across the 50 realisations and all the locations in space where the field was reconstructed. To study the variation across the number of sensors, we used N={9,16,25,36,49,64,81} and spread it uniformly across the field.

As we can see in the first top sub-figure in Figure 2, this is the situation in which the true spatial field process is very close to Gaussian as the g≈h≈0 settings was used and therefore with both *l* values only two curves are seen for the different spatial field reconstructions and they are basically almost indistinguishable. The difference reflects the very slight skew and kurtosis still present in the target warped spatial Gaussian process since *g* and *h* were not set exactly to 0. Basically in this scenario we see that the curve associated with the S-BLUE and the S-MAP coincide, as is the case with the S-MMSE and the GP curves. Furthermore all the estimators perform better as the number of sensors increase. Overall the difference in the reconstruction accuracy is negligible between the four estimators as is expected since the values of *g* and *h* are very close to 0 as is to be expected. This is simply a case study to illustrate that our spatial field reconstructions for the warped spatial GP model can recover a classical GP reconstruction if the target spatial field truly has this characterisation.

In the bottom subplot of Figure 2, we see that with some significant increase in *g* and *h* values that adds to the warped spatial GP skewness and kurtosis, then in this case the standard GP spatial field reconstructions degrade in accuracy. The classical GP estimate is unable to account for the significant non-linearities in the transform that induce the skewness and kurtosis in the spatial field, while the other estimators that we have derived for spatial field reconstruction specifically in the setting of warped spatial GP significantly outperform the classical GP reconstructions of comparative loss functional type, especially as the number of sensors increase. We also observe that the linearisation of the spatial field reconstruction provides an efficient estimator computationally compared to the MMSE estimator but the cost of this computational efficiency is a loss in accuracy due to the attempt to linearise locally the highly non-linear spatial field warping transforms. We see that it becomes particularly effective to use an efficient linear S-BLUE estimator in settings in which observation coverage is sparse and sensors are placed sparsely in the spatial region of interest. In these cases we show that the S-BLUE estimator performs comparatively with the more computationally expensive MMSE estimators.

**Figure 2 entropy-23-01323-f002:**
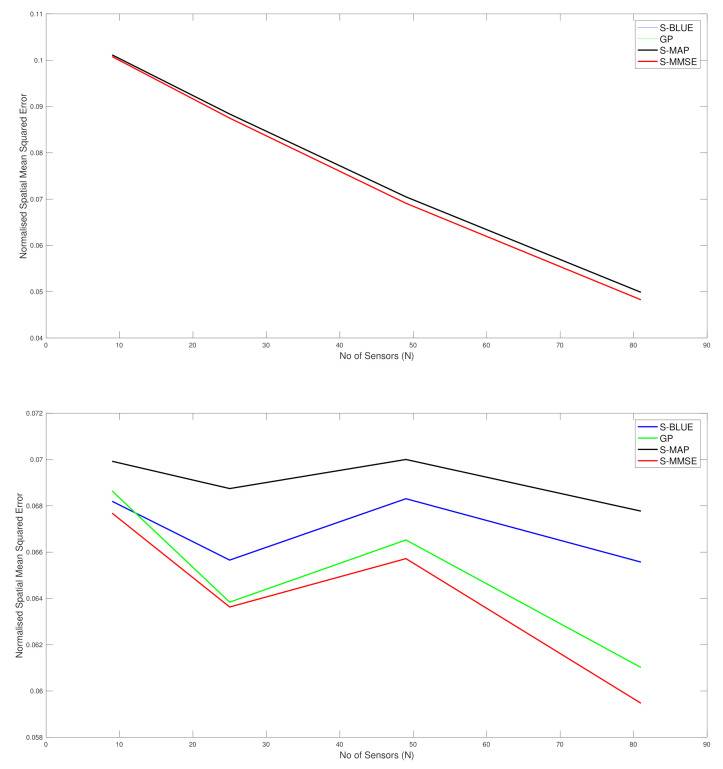
Plots of comparison of the N-MSE as a function of number of sensors (*N*). Top Subplot: g=0.001, h=0.001, l=0.5. Bottom Subplot: g=1.2, h=0.2, l=0.5.

### 5.3. Scenario 2: Spatial Exceedance on Synthetic Data

In the exceedance examples, we characterize the spatial region of exceedance estimator (Proposition 8) and its left tail equivalent, and the region of exceedance (Proposition 9). We treat these problems as a detection problem, since we obtain a binary estimate at each location. We characterize such performance via plots of the Receiver Operating Characteristic (ROC) curves associated with different values of *N*. In these illustrations we use α=0.1 and the threshold λ is varied from the minimum value of the realisations of the Tukey g-and-h field to the maximum value of the realisations. Once the predictive distribution is obtained at the target locations, we calculate the binary estimate of exceedance at each target location and finally to obtain the required probabilities of detection and false alarms, we collate the proportion of false alarms and correct detection at all locations and get the probabilities of false alarms and detection.

From the ROC curves in Figure 3 we see that the Tukey g-and-h distribution in different cases outperforms the Gaussian process in all the scenarios created, with closest performance, in Figure 3 subplot a, corresponding to cases when the generated synthetic data is from settings of g and h parameters in which the data generating mechanism is closest to no skewness or kurtosis effects. The case of a distinct model with regard to a Gaussian model, where skewness and kurtosis is non-negligible, is then shown in Figure 3 subplot b.

Now Figure 4 contains the ROC curve corresponding to g=1.2, h=0.001 and l=1. In subfigure a of Figure 4 one can see a scenario case that is close again to the Gaussian setting, with the difference between the case in Figure 3 subplot a being purely the scale parameter of the covariance kernel function, *l*. Therefore, as expected we see very close performance in the ROC curve in this setting when comparing the Gaussian and Tukey g-and-h models. In Figure 4 subplot b, we see the case with high positive skew and almost zero kurtosis. We notice that the false alarm rate for the quantile exceedance is extremely low and specifically we defined the exceedance as the right tail exceedance. This may be due to the fact that the estimation of the Tukey g-and-h posterior, at all the target locations, depend on the data from the sensors which tend to be considerably positively skewed (g=1.2), without heavy tails. Since, the GP estimation involves the actual observations (without any transformation), the predictive distribution tends to have a mean with a high positive value and since there is the asymmetry and the lack of heavy tails, the GP also exhibits a reasonably low false alarm rate for the right tail exceedance that is comparable to the Tukey g-and-h process. However, for the same case if we calculate the left tail exceedance as characterized in Proposition 9, we notice that these ROC curves exhibit a higher range of false alarms and detection rate and we obtain the following figure.

In addition, we observed that when we considered the case g=−1.2, h=0.001, l=1 which is negatively skewed with 0 kurtosis, we noticed that the above trend is reversed, i.e., the right tail exceedance exhibited a larger false alarm rate and the left tail exceedance exhibited a very small rate of false alarms.

**Figure 4 entropy-23-01323-f004:**
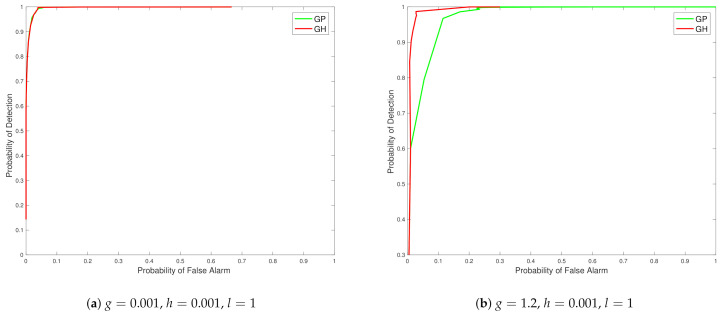
Plots of ROC curves comparing the Tukey g-and-h distribution to the Gaussian process for different values of *g* and *h*.

### 5.4. Scenario 3: Spatial Field Reconstruction on Real Spatial Sensor Data for Hourly Precipitation

In this section, we show results of spatial field reconstruction on the US precipitation dataset, available at https://www.ngdc.noaa.gov (accessed on 1 January 2020). This shows the hourly precipitation in digital data set DSI-3240, archived at the National Climatic Data Center (NCDC). It is obtained from US National Weather Service (NWS), Federal Aviation Administration (FAA), and cooperative observer stations in the United States of America, Puerto Rico, the US Virgin Islands, and various Pacific Islands. We extract data for one month (November 1994) from the sensors within the area shown in the map in Figure 5.

Here, to evaluate the performance of our estimators, we estimate the parameters from the US precipitation dataset corresponding to the data from the above locations. For both the Tukey g-and-h and the GP we use an underlying GP with zero mean and a squared exponential kernel where the length-scale and the variance are estimated appropriately. In addition, the Tukey g-and-h requires estimation of two additional parameters namely the skew parameter *g* and the kurtosis parameter *h*. After a maximum likelihood estimation in both cases, we obtain the estimates of the Tukey g-and-h to be $l_{gh}=0.2$, $\sigma_{gh}^{2}=1$, g=0.001 and h=0.2 and the estimates for the GP are $l_{gp}=0.1$ and $\sigma_{gp}^{2}=2.5$.

Additionally, we define a normalised version of the median of absolute deviations for the comparison of our estimates on the real data. It is defined as follows:$$\begin{array}{c} N\text{-}\mathrm{MAD}=median_{j\in J,i\in T}\frac{\left|\hat{Y}\left(x_{*}^{\left(ij\right)}\right)-Y\left(x^{\left(ij\right)}\right)\right|}{maxY-minY} \end{array}$$
where *J* is the number of spatial locations to regress and *T* is the number of trials of the Tukey g-and-h process simulation. In addition, $maxY=\max_{i,j}x^{\left(ij\right)}$ is the maximum value of the simulated process across all locations and trials and similarly $minY=\min_{i,j}x^{\left(ij\right)}$ is the minimum value of the simulated process.

**Figure 5 entropy-23-01323-f005:**
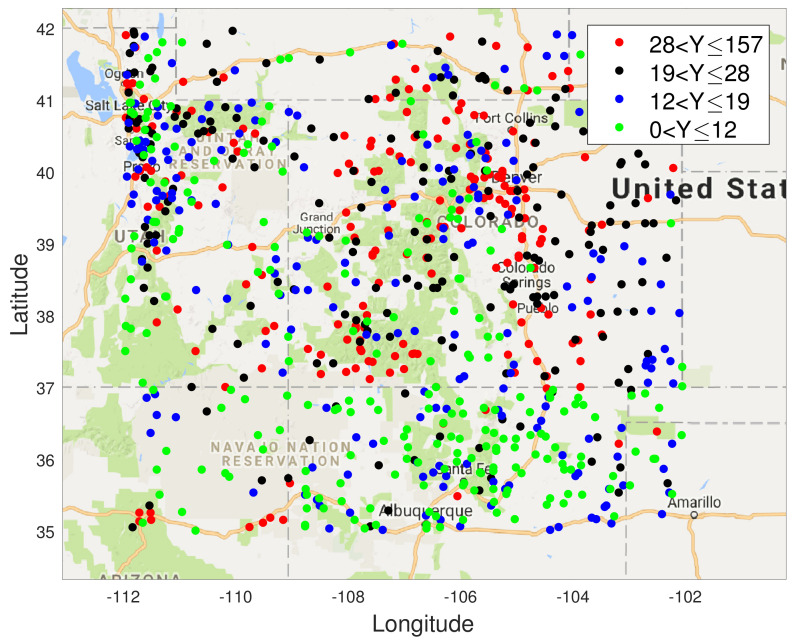
USA map with sensor locations measuring precipitation in 4 states showing the average precipitation for the month of November 1994 at 850 locations. The values (in mm) are colour coded according to the ranges shown.

In Figure 6 and Figure 7 we show results for a real data spatial field reconstructions for the S-Blue, GP, S-MAP and S-MMSE results. In Figure 7, we see that the ratio
$$R_{*}=\frac{N-MSE\;of\;the\;BLUE\;estimate\;at\;*}{N-MSE\;of\;the\;GP\;estimate\;at\;*},$$
in the majority (80%) of the test locations is less than or equal to 1, which means that the S-BLUE estimator performs better than the GP at those locations. The light blue markers indicated the locations where this ratio is between 1 and 5, which is around 11% of the test data. Similarly the dark blue markers and the black markers represent the small percentage of the locations where the GP performs better than the S-BLUE. We notice that these locations are generally concentrated in the corners of the grid, which might explain the presence of a few outliers as the main factor that affects the estimates is the covariance between the test location and the sensor locations. We can see from Figure 6, any estimator of the Tukey g-and-h process performs much better than the GP. Particularly in both cases the S-BLUE estimate performs the best in this case and this maybe because in the real data we do not have multiple realisations of the data, thus to get a good estimate we should exploit the spatial information well and the S-BLUE estimate does that precisely, as it requires the spatial Tukey g-and-h covariance, which enables it to produce better estimates and these estimates improve as the number of sensor locations having data increase. Moreover, we notice that the N-MSE of the GP increases as the number of sensors increases and this shows that we are fitting the data with a model that is biased.

## 6. Conclusions

We study in detail various aspects regarding Tukey g-and-h processes from its construction and derivation of *n*-variate cross moments, which helped us develop three different estimators for spatial regression, the MMSE, S-BLUE and the S-MAP. We discuss their derivation through the predictive posterior distribution. In addition, we define the right tail and the left tail exceedance estimators using the quantile level sets of these processes. Finally, we test the performance of our estimators on a simulated data setting and the US precipitation dataset. We find that although the Gaussian process is a useful model, we show that it suffers when the data exhibits heavy-tailedness and/or skewness, where the Tukey g-and-h process is better suited and is also practically applicable. The flexibility of the construction and the estimation of the Tukey g-and-h process can enable us to incorporate even more complex spatial and temporal patterns through different kernels much like the Gaussian process whilst enabling a way to account for non-Gaussian behaviour which arises in many real world applications.

## Figures and Tables

**Figure 1 entropy-23-01323-f001:**
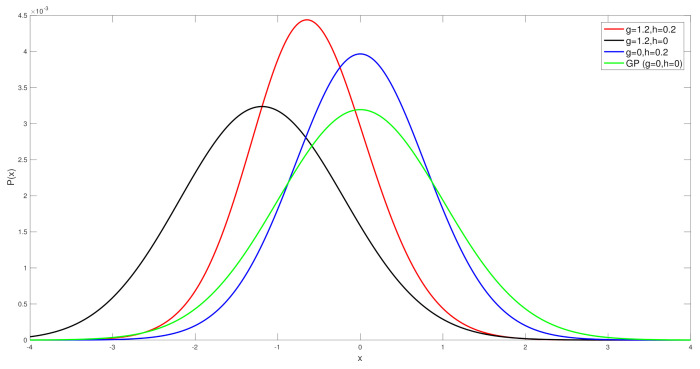
Comparison of the standard Gaussian distribution to the Tukey g-and-h distributions with values from $\left\{g=1.2,h=0.2,\mu =0,\sigma^{2}=1\right\}$.

**Figure 3 entropy-23-01323-f003:**
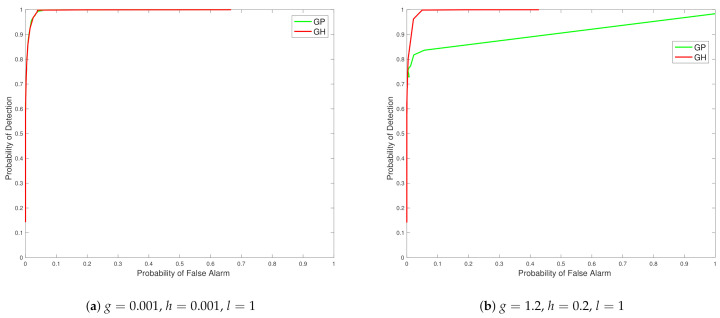
Plots of ROC curves comparing the Tukey g-and-h distribution to the Gaussian process for different values of *g* and *h*.

**Figure 6 entropy-23-01323-f006:**
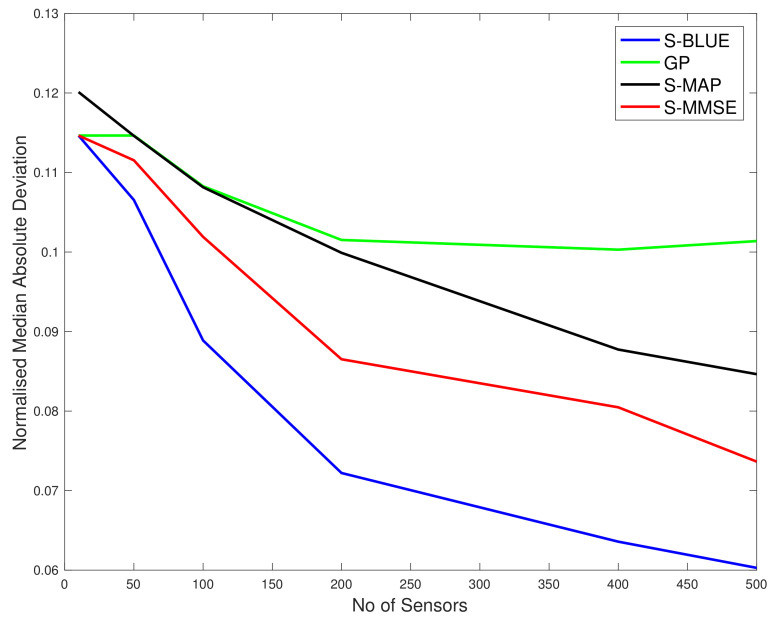
The N-MAD comparison between the various Tukey g-and-h estimators and the Gaussian process.

**Figure 7 entropy-23-01323-f007:**
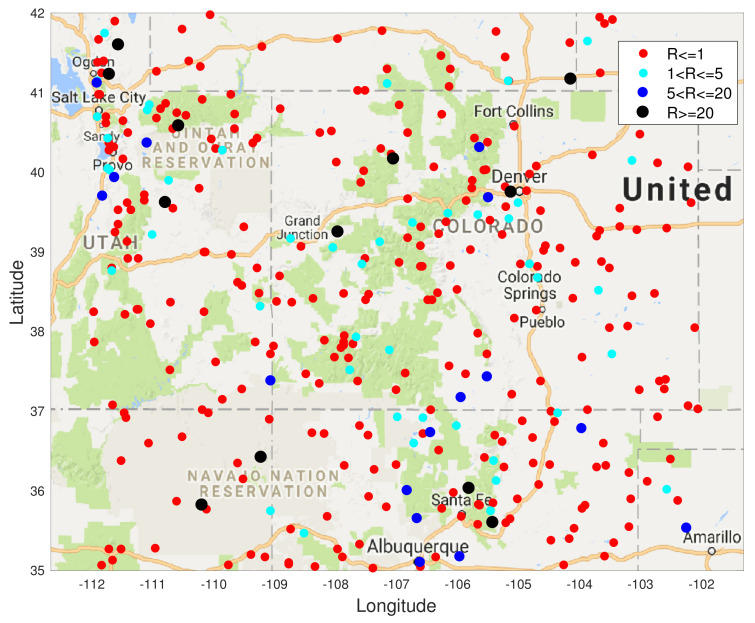
The ratio *R* of the N-MSE of the S-BLUE estimate and the GP estimate at each of the test locations when N=500, i.e., 500 locations were used as the training set.

## Data Availability

All real data utilised is available at the cited source location.

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
