# Peer review of "Spatial Warped Gaussian Processes: Estimation and Efficient Field Reconstruction"

_entropy, 2021, doi:10.3390/e23101323_

Round 1

Reviewer 1 Report

In the Reviewer’s opinion, this is a very interesting work by presenting a warped Gaussian processes for non-Gaussian applications. Experimental results show the effectiveness of the proposed models.

I have the following suggestions:

1) Non-Gaussian model is very useful for radar clutter or nonhomogeneous clutter [1-2]. It’s better to point out these potential applications in the introduction part.

[1] "Target Detection Within Nonhomogeneous Clutter Via Total Bregman Divergence-Based Matrix Information Geometry Detectors," in IEEE Transactions on Signal Processing, vol. 69, pp. 4326-4340, 2021, doi: 10.1109/TSP.2021.3095725.

[2] "Compound-Gaussian Clutter Modeling With an Inverse Gaussian Texture Distribution," in IEEE Signal Processing Letters, vol. 19, no. 12, pp. 876-879, Dec. 2012, doi: 10.1109/LSP.2012.2221698.

2) To model the radar clutter using the proposed warped Gaussian processes in the experimental part or add these applications in your future work.

Author Response

We are pleased that the reviewers found our work interesting. In response to the suggested minor updates:

  1. we have included the two references mentioned by the reviewer into the introduction of the manuscript.
  2.  We have decided not to implement new examples at this stage, the proposed example is interesting indeed for future work, but would not fit well the scope of the current manuscript in our opinion. We trust you will appreciate this given your wording suggesting yourself that this would be suitable for future work. We thank you for this suggestion.

best wishes, on behalf of co-authors

Prof. Peters

Reviewer 2 Report

This paper studies different aspects of Tukey g-and-h processes. Then, the authors develops different estimators for spatial regression. The topic that the authors try to address is important and very interesting.

The paper is well written and comes up with enough contributions. The authors should improve the English quality of the paper and try to minimise the amount of appendices (they may be embedded in the main text for a better reading).

Author Response

Dear reviewer,

We thank you for your time and reading of the manuscript. We are pleased that you found the work interesting to review and we thank you for your positive review.

In response:

  1. The authors should improve the English quality of the paper

We have been through the paper once more to further refine the English and make minor updates. We also ran the paper through a grammar and spell-check once again.

  1. and try to minimise the amount of appendices (they may be embedded in the main text for a better reading).

At present all key results are stated in the manuscript, including all theoretical results. The appendix contains proofs of the mathematical statements made in the main manuscript. We trust you will agree this is a fairly standard and reasonable approach to adopt when proofs for some results are algebraic and lengthy, so as not to disrupt the flow of the main messages in the manuscript.

We thank you for your time and consideration.

best wishes on behalf of co-authors

Prof. Peters

Reviewer 3 Report

The results in this paper seem to correct mathematically as well as geometrically. In my opinion the results are new and interesting and can attract the researchers working in this field. On the basis of these comments, I recommend the paper for publication in the Entropy after improving the English of the paper.

Author Response

Dear reviewer,

We thank you for your time and reading of the manuscript. We are pleased that you found the work interesting to review and we thank you for your positive review. In response:

  1. review the English quality of the paper

We have been through the paper once more to further refine the English and make minor updates. We also ran the paper through a grammar and spell-check once again.

We thank you for your time and consideration.

best wishes on behalf of co-authors

Prof. Peters

Round 2

Reviewer 1 Report

We recommend to publish this paper in the current version.